# Network-based diffusion analysis reveals context-specific dominance of dance communication in foraging honeybees

Matthew J. Hasenjager [1]*, William Hoppitt[1] & Ellouise Leadbeater [1]

The honeybee (Apis mellifera) dance communication system is a marvel of collective behaviour, but the added value it brings to colony foraging efficiency is poorly understood. In temperate environments, preventing communication of foraging locations rarely decreases colony food intake, potentially because simultaneous transmission of olfactory information also plays a major role in foraging. Here, we employ social network analyses that quantify information flow across multiple temporally varying networks (each representing a different interaction type) to evaluate the relative contributions of dance communication and hive-based olfactory information transfer to honeybee recruitment events. We show that virtually all successful recruits to novel locations rely upon dance information rather than olfactory cues that could otherwise guide them to the same resource. Conversely, during reactivation to known sites, dances are relatively less important, as foragers are primarily guided by olfactory information. By disentangling the contributions of multiple information networks, the contexts in which dance communication truly matters amid a complex system full of redundancy can now be identified.

[1] Department of Biological Sciences, Royal Holloway University of London, Egham TW20 0EX, UK. *email: matthew.hasenjager@rhul.ac.uk

During a waggle dance, a honeybee forager provides location information about a high-quality resource (e.g., foraging or nest sites) by producing multiple waggle runs, the duration and orientation of which correspond to spatial coordinates in the field[1]. That other foragers can use this spatial information to locate the indicated site[2,3] remains one of the most astounding discoveries of the last century within the field of animal behaviour[4]. Yet efforts to quantify the value that this spatial communication system brings to a colony's foraging operation have generally failed to find clear net benefits, repeatedly concluding that rendering the dance's directional component meaningless only rarely compromises colony foraging efficiency[5–9]. Concurrent work has emphasised that even for undisrupted dances, the spatial information provided is often of secondary importance to the dance's role as a trigger of navigational memories in dance followers[10–14], or to the olfactory information about profitable food sources that dancers provide through trophallactic nectar donations and scents carried on their bodies[10,13–18]. Since dance communication is integral to nest-site selection in honeybees[19,20], these findings have led to the suggestion that the role of dancing in foraging could in fact be a secondary one that is less critical than commonly supposed[21]. However, establishing whether foraging bees are responding to dance information, to alternative recruitment mechanisms, or to a combination of these, is challenging given that all information sources are typically available simultaneously within the hive.

To overcome this difficulty, we use network-based diffusion analysis (NBDA)[22,23] to tease apart the relative importance of dance-based communication networks, in relation to individual search and olfactory information transfer, in guiding honeybees to both novel and known foraging sites. NBDA has enabled in-depth study of social transmission processes across a range of vertebrate taxa[24–29]. Its core assumption is that if a learnt behaviour or piece of information—such as discovery of a novel foraging site—spreads via social transmission, then its diffusion will follow a social network[30]. Here, we extend this approach to allow for the simultaneous inclusion of multiple, time-varying social networks in order to identify the key information pathways amongst honeybee foragers.

By training cohorts of bees to novel feeders, we mimic natural foraging events and observe the in-hive interactions that follow in order to build alternative social networks based on dance interactions, trophallactic nectar donations and antennation of returning foragers. All three interaction types are known to motivate honeybees to search out known or novel sites in the field[1–3,11,13,31–33]. However, only the dance additionally provides navigational information[2,3], whereas trophallaxis and antennation (both of which can also occur during dance following) lead to learning about the scent of the target foraging site[15,16,18]. We estimate the power of each network to predict the order of arrival at the feeders, and find that in a context mimicking natural depletion of one patch of a flower species followed by discovery of

another, waggle dance communication is the dominant mode of transmission. Neither of the two main olfactory information pathways—antennation and trophallaxis—are important for discovering foraging sites in this particular context. Conversely, all three pathways contribute to motivating temporarily unemployed foragers to resume collecting from a known, familiarly scented site, but antennation is especially important in this regard. By revealing how alternative information pathways combine to shape behaviour, NBDA offers a promising approach for identifying the contexts in which dance communication matters most in driving honeybees to food.

## Results

**Following dances is crucial for recruitment to novel sites.** Across four trials, each conducted on a separate colony (Table 1), 190 marked bees were trained to one of two identical feeders (18–30 foragers per feeder per trial; colonies tested sequentially; Fig. 1). On the day immediately preceding testing, both feeders provided identically scented sucrose to allow the formation of food-odour associations. Then, for each colony, one feeder (hereafter EMPTY feeder) ceased to provide sucrose on the test day that immediately followed this odour presentation. Bees that had been trained to this feeder, on discovering that it had become unrewarding, were thus unemployed and available to be recruited to the other (hereafter FULL) scented feeder. As foragers were uniquely identifiable, we could be sure potential recruits had never before visited the novel-to-them FULL feeder.

During a trial, foragers returning from the FULL feeder: (i) transmitted its location by producing waggle dances; (ii) provided scented nectar samples through trophallaxis; and (iii) bore food-associated scents on their bodies. Each of these interaction types are known to motivate bees to forage[1,11,13,33] and could have informed potential recruits that despite the decline of their own familiar feeder, a new resource patch of the same type had become available. While waggle dances additionally transmit spatial information, exposure to scent can lead foragers to search out matching sites, particularly over relatively short foraging distances (≤~300 m) comparable to the present study[2,31,32]. We therefore built separate networks that quantified dance-following interactions, trophallactic nectar exchange and antennal contacts between potential recruits and foragers collecting from the FULL feeder (Fig. 1e–g). We then used NBDA to assess the relative influence of these networks over the order in which recruits discovered the FULL feeder. For example, if the dance pathway were of primary importance, we would expect the order to closely follow the dance network, but not the antennation or trophallaxis networks.

Across the 4 trials, 56 individuals were recruited to the FULL feeders. The level of support received by each network or predictor variable in the NBDA is given by its summed Akaike weight ($\Sigma w_i$), which indicates the probability that the best

**Table 1 Overview of experimental trials.**

| Colony | Start of training | Trial date | Trained individuals (FULL/EMPTY) | Reactivated to FULL feeder | Recruits to FULL feeder |
|---|---|---|---|---|---|
| 1 | August 23 | August 29 | 25/24 | 25 | 15 |
| 2 | September 1 | September 6 | 30/29 | 30 | 34[a] |
| 3 | September 14 | September 23 | 18/18 | 16 | 4 |
| 4 | September 25 | October 5 | 20/26 | 20 | 3 |

Overview of trial dates, the number of individuals trained to each feeder prior to each trial, and the number of individuals that were reactivated and recruited to the FULL feeder during the trials. Reactivation involved the FULL group returning to their familiar FULL feeder, while recruitment involved the EMPTY group arriving at the FULL feeder, which they had never previously visited.
[a]Occasionally, marked foragers stopped visiting their respective feeders during the first or second day of training when feeders were still distantly located (>100 m) from their final destination. Such individuals were therefore naïve regarding the FULL feeder's location. Ten such individuals were recruited during the second trial, resulting in more recruits than there were individuals trained to the EMPTY feeder.

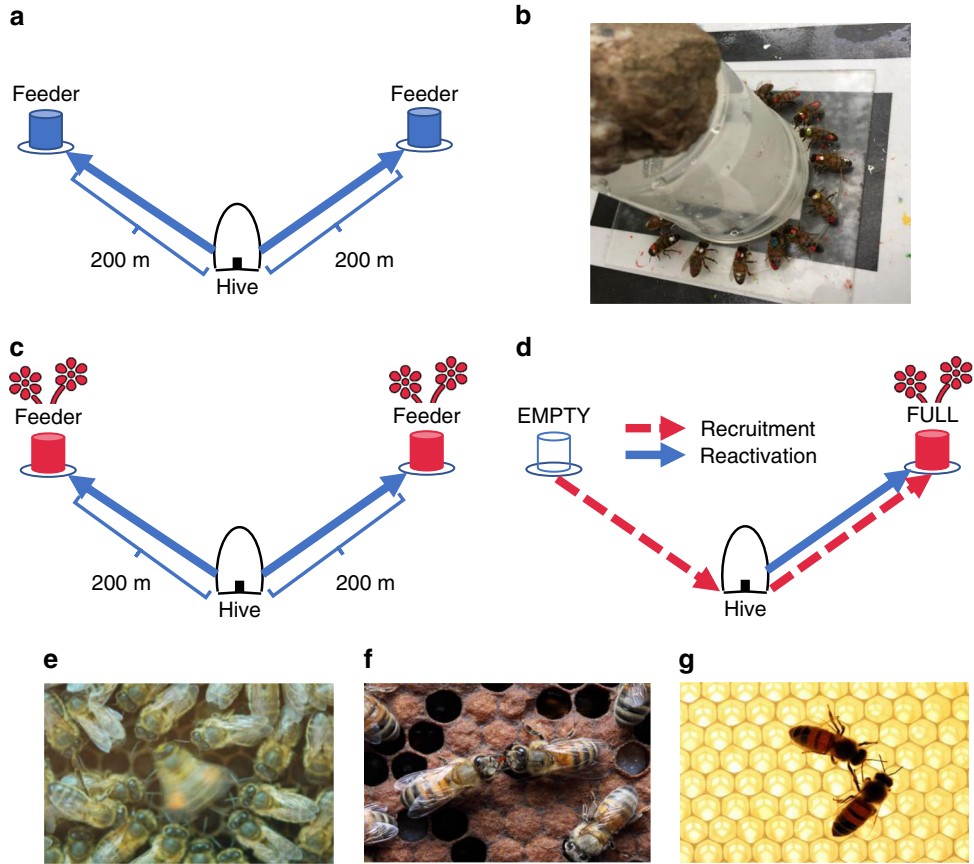

**Fig. 1 Experimental design. a** For each trial, 18–30 foragers from a single colony were trained to a feeder that provided unscented sucrose solution. The feeder was gradually moved to a location 200 m from the hive. A second group from the same colony was simultaneously trained to an identical feeder, with an angular separation of ~110°, using the same methods. **b** Each forager was given an individually specific enamel paint marking upon its first arrival at a feeder. **c** On the morning before the trial, both feeders provided 2 M sucrose solution identically scented with an essential oil for 1 h (indicated in red), thereby allowing individuals to associate the provided scent with their foraging site. **d** During the trial, the FULL feeder provided 2 M sucrose solution identically scented as the previous morning, whereas the EMPTY feeder was left unfilled. Bees in the FULL group could be reactivated to their familiar FULL feeder (reactivation NBDA; indicated by the solid blue line), whereas bees in the EMPTY group, upon discovering that their familiar feeder was now empty, became available for recruitment to the FULL feeder (recruitment NBDA; indicated by the dashed red line); see Supplementary Note 1 for an NBDA of the EMPTY group being reactivated to the EMPTY feeder. We recorded the order of both reactivation events and recruitment events within the same trial. Social networks were constructed that quantified bouts of dance-following (**e**), trophallaxis (**f**) and antennation (**g**) between marked individuals and foragers returning from the FULL feeder. Image **e**: taken by Christoph Grüter, used by permission. Images **f** and **g**: © Alexander Wild, used by permission.

predictive model contains that variable[34]. The network quantifying dance-following interactions received unequivocal and overwhelming support ($\Sigma w_i > 0.999$; Table 2). In addition, the number of waggle runs followed for the FULL feeder predicted arrival order during recruitment more accurately than the total duration of contact with dancing bees (Supplementary Table 1). This suggests that it was specifically the spatial information encoded in each waggle run[1] that was of key importance, and that by following more runs, more accurate location information could be obtained[35,36]. In contrast, both the antennal contact and trophallaxis networks received little support, regardless of whether network connections were weighted according to the number of interactions or their total duration (all $\Sigma w_i = 0.29$; Table 2). In other words, despite an extensive body of evidence showing that olfactory information can lead unemployed bees to novel foraging sites[2,31,32,37,38], we found little evidence that it does, at least within this particular context where foragers sought out a new patch of a familiar flower species.

Here, network effects were modelled as independently influencing the rate of feeder discovery, but it may be that recruitment order depended on interactions between different

**Table 2 Support ($\Sigma w_i$) for social transmission pathways in the recruitment NBDA.**

| Social network[a] | $\Sigma w_i$ |
|---|---|
| Dance-following network | >0.999 |
| Trophallaxis network (duration) | 0.290 |
| Trophallaxis network (number of interactions) | 0.290 |
| Antennal contact network (duration) | 0.290 |
| Antennal contact network (number of interactions) | 0.290 |
| Homogeneous network[b] | <0.001 |
| No network (asocial learning only)[c] | <0.001 |

[a]Variants were considered for both the trophallaxis and antennal contact networks in which network connections were weighted either by total interaction duration (s) or according to the number of separate interactions (regardless of their duration) between two individuals.
[b]The homogeneous network included a connection with a strength of 1 between every forager upon its first return to the hive from the FULL feeder and all potential recruits that had not yet departed the hive for the FULL feeder. Support for this network would have indicated that none of the measured social networks provided a sufficient approximation for the true pathway(s) of social transmission[41].
[c]Asocial models assume that feeder discovery occurred through independent search alone, without reliance on social information. Note that asocial learning is also assumed to operate within NBDA models that include a social transmission component[23].

information pathways—e.g., if receiving nectar samples modulated attentiveness to dances. However, we found little evidence supporting interactive effects between the dance-following network and either the trophallaxis or antennal contact networks (best supported model without interactions: $w_i = 0.492$; all alternative models: $w_i < 0.161$; Supplementary Table 2). Note also that there were substantially fewer successful recruits in the last two trials (Table 1), coinciding with typically increased availability of natural food sources (which bees usually favour over feeders) in early autumn in southern England—esp. ivy (*Hedera* spp.)[39,40] (Supplementary Table 3). Nevertheless, an NBDA of the first two trials alone yielded the same result as the NBDA that included all four trials: only the dance network predicted recruitment order (Supplementary Tables 4–6).

To rule out alternative explanations for the observed order of arrival at the feeder, we included two additional sets of models in our analysis. In principle, foragers could have discovered the FULL feeder solely through independent search without relying on socially transmitted information. However, models that allowed for only such asocial learning of the feeder's location received virtually no support ($\Sigma w_i < 0.001$; Table 2). Furthermore, that transmission specifically followed the dance network is shown by the absence of support for models of homogeneous social transmission ($\Sigma w_i < 0.001$; Table 2)—that is, equal transmission rates among all dyads regardless of any interactions that were observed to occur between them. If the homogeneous network had been supported, this would have suggested that the actual social transmission pathways differed substantially from those approximated by our measured networks[41].

Model-averaged estimates and 95% CI for social transmission rates for the recruitment diffusion are presented in Table 3. NBDA estimates the social transmission rate, $s$, as the rate at which an individual acquires information per unit of network connection with informed individuals, relative to the rate of asocial learning[22,23]. In other words, $s$ quantifies the extent to which observing or interacting with knowledgeable individuals accelerates learning—e.g., of a novel behavioural pattern[25,26,28] or foraging patch location[27,29]—beyond what is expected through individual exploration. Here, social transmission rates were estimated relative to the baseline feeder discovery rate for the average individual in trial 1. This means that after following just a single waggle run for the feeder during a bout of dance following, the average potential recruit in the first trial was expected to discover the feeder 10.4 ($s + 1$) times sooner than if it had not followed any (Table 3).

Following[25], we can convert $s$ into the estimated proportion of recruitment events that occurred as a result of each transmission pathway while accounting for the contribution of alternative pathways. Of the 56 recruitment events, 93.4% (95% CI: 76.4–97.9%) were explained by dance-following interactions. Conversely, far fewer recruitment events were explained by either trophallaxis (0.1%; 95% CI: 0–7.6%) or antennal contacts (4.6%; 95% CI: 0–21.6%). These results indicate that following dances was crucial for successful recruitment to the novel foraging site (Supplementary Fig. 1). Despite the widespread occurrence of trophallaxis and antennation in the hive, and reported successful foraging in the absence of location communication[5–9,32,42], odour-guided search on its own contributed almost nothing to recruitment within the timeframe and conditions of this experiment.

Two additional predictors were also considered in the recruitment NBDA: trial (synonymous with colony; Table 1) and the number of times prospective recruits returned to inspect their familiar, now-empty feeder. The NBDA approach allows for examination of how these variables impact both the rate of asocial learning through independent search, and of social transmission through the interaction networks. Though several factors co-varied across trials that could have influenced recruitment, such as colony genetic make-up or weather conditions, there was little evidence of variation in learning rates across trials (asocial effect: $\Sigma w_i = 0.119$; social effect: $\Sigma w_i = 0.171$; Supplementary Table 7). However, individuals that visited the EMPTY feeder more persistently were slower to find the FULL feeder, though not simply because individuals were unable to simultaneously inspect the EMPTY feeder and search for the FULL one (asocial effect: $\Sigma w_i = 0.25$; Supplementary Table 7). Rather, repeatedly visiting the EMPTY feeder translated into lower social transmission rates (social effect: $\Sigma w_i = 0.947$; $\beta = -0.25$; Supplementary Table 7). This indicates that though foragers often followed dances advertising the FULL feeder prior to these return visits, they did not necessarily attempt to acquire its spatial coordinates during these bouts of dance following (see also refs. [12,13,43]). That variation in persistence to previously rewarding sites impacted the likelihood of recruitment may aid honeybee colonies in exploiting newly discovered resources without completely abandoning depleted sites that may yet yield rewards in the future[44].

**Reactivation relies on multiple information pathways.** Following dances in order to locate a new foraging site is thought to account for a relatively low proportion of all dance interactions in the hive (estimated at 12–25%)[12]. The remainder (75–88%) take place in the context of motivating continued inspection of a known food source, or reactivating to it following an interruption (e.g., by inclement weather or nightfall). Such an interruption was mimicked by our experimental design: on the day before each trial, the FULL feeder offered high-quality sucrose for 1 h and then was removed (Fig. 1c), halting foraging for the remainder of the day. It then became available once more at the start of the trial (Fig. 1d). This set-up thus allowed us to investigate within the same trial, the role of each interaction network not only in recruiting bees that had been trained to the EMPTY feeder, but in reactivating those bees that had been trained to the FULL one.

Following the same protocols described previously, we built networks that quantified interactions between foragers returning from the FULL feeder and individuals that had been trained to the FULL feeder, but had yet to reactivate. Across all four 2 h trials, 91 foragers reactivated to the FULL feeder (Table 1). We restricted our analysis to those events for which individuals' in-hive interactions prior to reactivation were observed ($n = 67$). For example, some individuals were already present (and presumably waiting) when the FULL feeder was put in place, prohibiting us from assessing the relative importance of personal information use vs. in-hive interactions over those reactivation decisions.

---

**Table 3 Estimated rates of social transmission ($s$) in the recruitment NBDA.**

| Social transmission parameter, $s$ | Model-averaged estimate (95% CI)[a] |
|---|---|
| Dance-following $s$ | 9.41 (1.64–174.60) |
| Trophallaxis (duration) $s$ | 0.008 (0–1.25) |
| Antennal contact (number of interactions) $s$ | 0.21 (0–1.25) |

[a]Model-averaged estimates were obtained using the variants specified above for the trophallaxis and antennal contact networks due to these variants receiving more support ($\Sigma w_i$) than their alternatives; see Methods for further details. Confidence intervals were obtained using profile likelihood techniques[67] from the highest ranked model that included a given network. The profile likelihood for $s_{antennal\ contact}$ from the highest ranked model ($\Delta$AICc = 2.235) that included this parameter showed evidence of practical nonidentifiability[69]. As such, confidence intervals for $s_{antennal\ contact}$ were obtained instead from the next highest ranked model, which received virtually the same support ($\Delta$AICc = 2.235). This latter model was also the highest ranked model to include the trophallaxis network, and constrained $s_{trophallaxis}$ and $s_{antennal\ contact}$ to be equal.

**Table 4 Support ($\Sigma w_i$) for social transmission pathways in the FULL feeder reactivation NBDA.**

| Social network[a] | $\Sigma w_i$ |
|---|---|
| Dance-following network | 0.848 |
| Trophallaxis network (duration) | 0.07 |
| Trophallaxis network (number of interactions) | 0.781 |
| Antennal contact network (duration) | 0.05 |
| Antennal contact network (number of interactions) | 0.836 |
| Homogeneous network[b] | <0.001 |
| No network (asocial reactivation only)[c] | <0.001 |

[a]Variants were considered for both the trophallaxis and antennal contact networks in which network connections were weighted either by total interaction duration (s) or according to the number of separate interactions (regardless of their duration) between two individuals.
[b]The homogeneous network included a connection with a strength of 1 between every forager upon its first return to the hive from the FULL feeder and all individuals that had not yet reactivated nor departed the hive for the FULL feeder. Support for this network would have indicated that none of the measured social networks provided a sufficient approximation for the true pathway(s) of social transmission[41].
[c]Asocial models assume that reactivation was entirely self-initiated, without relying on social information. Note that asocial reactivation is also assumed to operate within NBDA models that include a social transmission component[23].

**Table 5 Estimated rates of social transmission (s) in the FULL feeder reactivation NBDA.**

| Social transmission parameter, $s$ | Model-averaged estimate (95% CI)[a] |
|---|---|
| Dance-following $s$ | 5.72 (1.05–70.39) |
| Trophallaxis (number of interactions) $s$ | 1.29 (0.05–13.73) |
| Antennal contact (number of interactions) $s$ | 0.97 (0.05–13.73) |

[a]Model-averaged estimates were obtained using the variants specified above for the trophallaxis and antennal contact networks due to these variants receiving more support ($\Sigma w_i$) than their alternatives (Table 4); see Methods for further details. Confidence intervals were obtained using profile likelihood techniques[67] from the highest ranked model; this model constrained the estimates for $s_{trophallaxis}$ and $s_{antennal\ contact}$ to be equal.

The NBDA found strong evidence for a combined role of all three types of interaction in reactivating foragers (Table 4). Specifically, the order in which individuals first returned to the FULL feeder was predicted by the number of waggle runs followed for the feeder ($\Sigma w_i = 0.848$), along with the number of trophallactic ($\Sigma w_i = 0.781$) and antennal contacts ($\Sigma w_i = 0.836$). Although reactivation can be self-initiated without requiring stimulation from other foragers[11,12], models that included only asocial influences over reactivation received virtually no support ($\Sigma w_i < 0.001$; Table 4), as did models of homogeneous social transmission ($\Sigma w_i < 0.001$; Table 4). We again found no evidence indicating that the order of reactivation depended on interactive effects between these networks (best supported model without interactions: $w_i = 0.529$; all alternative models: $w_i < 0.17$; Supplementary Table 8).

Model-averaged estimates and 95% CI for social transmission rates are presented in Table 5. Far fewer (14.8%, 95% CI: 5.6–25.3%) reactivation events were explained by dance-following interactions than had been the case for recruitment events, while conversely, the number of events explained by olfactory pathways increased (trophallaxis: 10.5%, 95% CI: 0.7–33.1%; antennation: 37.7%, 95% CI: 6.5–73.7%). Most likely, resumption of foraging was motivated by detecting learnt food odours, which are known to trigger navigational memories associated with familiar sites[1,10,45]. In addition, an estimated 37% of reactivations occurred independently of these transmission pathways, reflecting personal information use[11,12] and/or responses to social cues not captured in our networks (e.g., detecting food-associated odours on unmarked bees that had previously interacted with foragers collecting from the FULL feeder). Both asocial reactivation rates and the strength of social influences over reactivation varied across trials (asocial effect: $\Sigma w_i = 0.994$; social effect: $\Sigma w_i = 0.957$; Supplementary Table 9), potentially due to unidentified environmental influences (e.g., weather conditions). Consistent with previous work[11,12], we also found that foragers that had spent more days foraging at the FULL feeder prior to the trial were faster to reactivate, independent of any social influences (asocial effect: $\Sigma w_i = 0.923$; $\beta = 1.22$; Supplementary Table 9). In Supplementary Note 1, we provide the results of the complementary analysis focusing on reactivation to the EMPTY feeder; briefly, trophallaxis and antennation were responsible for the majority of reactivation events to the EMPTY feeder, with dance communication playing only a limited role (Supplementary Tables 10–13).

## Discussion

That odour-based and dance-based recruitment operate in parallel within honeybee hives has been recognised for many years[1], and was central to a major challenge to Karl von Frisch's original description of the dance recruitment system that played out in the latter half of the 20th century[37,46]. While James Gould convincingly dispelled claims that odour-based recruitment could be the only means by which dance-followers located foraging sites—performing an *experimentum crucis* in which followers were recruited to sites that had never been visited by the dancers that indicated them—he nonetheless concluded that "the inherently 'spectacular' nature of the dance language may have helped to emphasise it out of proportion to its actual place in the ecology and dynamics of foraging", and that "only further work can establish whether the dance-language is common or rare under normal circumstances"[2]. We have shown that NBDA offers a means to meet this challenge by identifying those contexts in which dances are the major driver of arrivals at foraging sites. In our experiment, dance following was the major elicitor of discovery of new patches of a target flower species. Conversely, although bees also follow dances before reactivating to known food sites, trophallaxis events explained almost the same number of feeder arrivals during reactivation as dance following, and antennation explained over twice as many. By disentangling the effects of alternative social networks, the NBDA approach thus offers a broadly applicable means of identifying the key social learning pathways that operate simultaneously within freely interacting animal groups.

That dance-conveyed information may simultaneously guide followers engaged in distinct foraging activities—a property known as pluripotentiality in systems science[47,48]—may go some way towards explaining why studies have often failed to detect clear benefits to this communication system[5–9]. The costs of producing spatial information under ecological contexts in which it is less valuable may be offset by the benefits gained through the dance's motivational, reactivation and reaffirmation functions[1,12,13,49]. At the same time, because reactivation is also elicited by antennation and trophallaxis, these other transmission pathways may aid in maintaining the dance communication system through such overlapping functionality (i.e., degeneracy[47,48,50]). In other words, responses to selection pressure favouring the transmission of spatial information may be enhanced, as the dance's reactivation function can be fulfilled through alternative information networks. Indeed, honeybee colonies may only gain weight during brief periods of intense nectar flow each year[51], during which time the ability of the dance to direct foragers towards highly profitable foraging sites may be crucial for colony survival.

While our results confirm the key role of the dance's spatial information during recruitment to novel sites, they also suggest that previous studies may have underestimated the importance of

non-dance interactions for honeybee foraging. For instance, foragers that leave the hive without having followed any dances have often been assumed to be scouting for novel resources[12,14,52]. Yet many of these bees are likely to have engaged in olfactory interactions in the hive, which could motivate them to either revisit known sites[10,13,45] or search for a particular flower species[2,31,33]. Trophallactic exchange has long been considered potentially key to this process[15,16]. More recently, it has been demonstrated that even without experiencing a sucrose reward, simply detecting floral scents during antennation can be sufficient for honeybees to learn food-odour pairings[18], and to preferentially select food sources bearing that scent when encountered in the field[33]. Our findings support the importance of this network as a driver of reactivation, and reveal that under certain circumstances, information transferred during such interactions can actually be of greater importance for organising honeybee collective foraging than that gained from following waggle dances. Intriguingly, as antennation occurs within all social bees (including honeybees, bumblebees and stingless bees), similar transmission pathways may be present across species and play a similar role in coordinating forager efforts.

Although detection of familiar odours seems the most likely explanation for our finding that antennal contact was the single most important driver of reactivation, it is also possible that simply coming into contact with employed foragers was sufficient to elicit reactivation[49]. Future work should therefore explicitly examine how food odour familiarity influences the relative importance of these alternative information networks. In addition, despite its well-established ability to facilitate olfactory learning[15,16] and rapidly distribute collected nectar through the hive[53,54], we found that trophallaxis contributed little towards forager recruitment. The experimental design used here may have limited the importance of trophallaxis for successful recruitment, as recruits had learned the food odour prior to the trial. Thus, after being guided to the feeder's general proximity via the dance, it is likely that they could simply home in on the familiar scent without having needed to experience it first in the hive[38]. In contrast, during recruitment to an unfamiliarly scented resource, trophallaxis may enable foragers to rapidly learn the novel odour, which could then be used in tandem with spatial communication to successfully locate the advertised site. It has also been hypothesised that brief trophallactic contacts facilitate assessment of the relative quality of available resources[55–57], such that deteriorating foraging options can be abandoned more rapidly.

Perhaps the most pressing question, however, is whether honeybees do in fact rely more heavily on dance information when resources are scarce and patchily distributed, as has previously been suggested[5–7,9,58,59]. Identifying the circumstances under which honeybees use information gained through dance communication is crucial if we are to understand the selective pressures that may have shaped its evolution. Automated behavioural tracking technologies are rapidly advancing to the point that researchers will soon be able to construct detailed interaction networks involving multiple interaction types and encompassing the entire hive[60–63]. Combining such data with NBDA may finally provide the means of answering one of the most enduring and fascinating mysteries in the field of animal behaviour: what is the adaptive value of honeybee dance communication, given that this system has failed to evolve in even a single other social insect group?

## Methods

**Colony housing.** Trials were conducted on the campus of Royal Holloway University of London from August–October 2017 (Table 1). At this time of year, natural food sources are relatively scarce compared to spring and early summer[39], thereby ensuring honeybees' interest in the provided sucrose feeders.

Four queen-right honeybee *Apis mellifera* colonies were housed indoors within three-frame glass-walled observation hives, with unrestricted tunnel-based access to the outdoors. Each colony contained 2000–3000 workers, brood and reserves of pollen and honey.

**Training procedure.** Our experimental design followed a dual-feeder set-up, whereby one cohort of bees (FULL group) were trained to a feeder that offered rewards throughout training and testing, while a second (EMPTY group) visited a feeder that became unrewarding after training (Fig. 1). During testing, the EMPTY group thus provided a pool of marked, motivated bees that were unemployed and amenable to recruitment within the timeframe of our experiment. Working with one colony at a time ($n = 4$ colonies), we trained the two cohorts (18–30 individuals per group) to their respective feeders over 4–9 days (depending upon weather conditions) following standard protocols[1,51]. During training, newly arrived unmarked foragers were individually marked at the feeder with enamel paints (Humbrol®; Fig. 1b). Subsequent in-hive visual observation served to confirm whether newly marked individuals belonged to the focal colony. Any individuals not observed in the hive were captured upon returning to the feeder and later frozen, thereby preventing non-focal colonies from interfering with or outcompeting our focal cohorts at the feeders.

At the end of training, feeders were located 200 m from the hive with an angular separation of ~110° as observed from the hive (Fig. 1a). Any individuals that switched between feeders during training were captured, so trained foragers almost certainly knew of only one feeder location following training. Four sites were used across the study, with each serving once as the location that individuals were recruited from (EMPTY feeder) and once as the location individuals were recruited to (FULL feeder). As the length of time taken to complete training varied across trials (Table 1), and because individuals were recruited to their feeders at different times during that process, we used unscented sucrose during training until the final pre-trial day, to limit among-individual variation in exposure to the scented sucrose used during the odour treatment and trial.

As exposure to familiar food odours may be important for forager reactivation and recruitment[10,13,45], feeders provided scented 2 M sucrose solution for 1 h (~1000–1100 h) on the day prior to the trial (Fig. 1c). Following previous work[13,43,51], we mixed 50 μL essential oil per litre sucrose solution. During this period (hereafter the odour treatment), foragers made several collecting trips to their feeder (mean ± SD visits = 6.08 ± 3.06), thereby allowing individuals to associate the provided scent with their familiar feeding site. Honeybees can form long-term olfactory memories after as few as three exposures[64]. The same scent was provided at both feeders within a trial and a different scent was used for each colony (colony 1: peppermint; colony 2: geranium; colony 3: lavender; colony 4: orange). After 1 h, both feeders were removed for the remainder of the day.

**Trial procedure.** On the day following the odour treatment, both feeders were set up for 2 h (~0930–1130 h). The FULL feeder supplied 2 M sucrose solution scented identically as during the odour treatment, while the EMPTY feeder was left unfilled (Fig. 1d). As both feeders had been equally profitable the day before, individuals trained to either feeder should have been equally motivated to return on the morning of the trial to assess their foraging site's continued profitability. Those bees returning from the EMPTY feeder, having learnt that it was no longer rewarding, were thus amenable to recruitment to the FULL feeder, which represented a new patch of the focal flower species. Recruitment could have been mediated either by individual search or by interactions (e.g., dance following, trophallaxis and antennation) with foragers collecting from the FULL feeder. Initial arrivals at the FULL feeder by members of the EMPTY cohort were the focus of the "Recruitment NBDA". At the same time, by recording initial arrivals at the same FULL feeder by bees in the FULL group, we also could examine reactivation of trained individuals to their familiar foraging site (reactivation NBDA). Note that we also analysed reactivation of the EMPTY group to the EMPTY feeder in an additional NBDA (see Supplementary Note 1). Trials lasted for 2 h, and all arrival times were verified by video (Canon Legria HF R806 digital camcorder). Any unmarked bees that arrived at the FULL feeder were captured by an observer before they could return to the hive.

In-hive interactions during each trial were filmed (Canon Legria HF R806 digital camcorder). Observation hives were constructed such that returning foragers remained on one side of the hive because a wooden ballast prevented easy crossing[51]. Each individual returning from the FULL feeder was followed for the complete duration of its hive stay, meaning we were able to obtain a near-complete record of all contacts with individuals knowledgeable about the FULL feeder (hereafter demonstrators). We recorded the following types of interaction between these demonstrators and other marked individuals: (a) dance following: an individual was oriented towards and within one antennal length of a demonstrator during the latter's performance of a waggle dance (Fig. 1e); (b) trophallaxis: an individual's extended proboscis contacted a demonstrator's mouthparts (Fig. 1f) and (c) antennal body contact: an individual's head was directed towards and within one antennal length of the demonstrator's body (Fig. 1g). By definition, antennal body contact occurred during any instance of dance-following and trophallaxis, but also occurred outside these contexts. The times at which each interaction began and ended were recorded, as were the number of waggle runs followed for each bout of dance following.

The NBDA did not include other sources of social information that can aid foragers in pinpointing a resource once in the field, such as forager-produced pheromones or visual/olfactory cues[65,66]. Since these cues were likely available to all bees once they reached the immediate vicinity of the food source, failing to include these potential transmission opportunities means that the estimated strength of social transmission through our focal networks is, if anything, likely to be conservative[30].

The experiments described here were conducted in accordance with guidelines established by the Research Ethics Committee of Royal Holloway University of London.

**Network-based diffusion analysis.** NBDA quantifies the rate at which naïve individuals first express a target behavioural pattern as a function of their connectedness to knowledgeable demonstrators[22,23]; for a mathematical description of the model, see Supplementary Note 2. The behaviours of interest in the current study were discovery of the FULL feeder's location by any individual trained to the EMPTY feeder (recruitment) and for those individuals trained to the FULL feeder, an individual's first return to its familiar feeder (reactivation); for both behaviours, individuals were assumed to become informed at the time at which they first landed on the feeder of interest. The core assumption underlying NBDA is that if a target behaviour is transmitted socially, then its diffusion is predicted to follow a social network reflecting social transmission opportunities[30]. The standard NBDA model incorporates a single social network, and so cannot determine whether transmission rates vary across different types of social connection. Here, we used a recently developed NBDA variant[27] that permits researchers to incorporate multiple social network types within a single model and thereby test whether social transmission rates vary across these different pathways.

The standard NBDA model also assumes social network structure remains static during a diffusion, an assumption that can be misleading when networks directly quantify social transmission opportunities. For instance, imagine three honeybees (A–C) during a recruitment trial, with individual A knowing the feeder's location. Individual B first follows six waggle runs produced by A before it discovers the novel feeder. Next, individual C follows ten waggle runs produced by A before it too finds the site. A static network would predict that C should arrive before B, whereas a dynamic network that takes the time course of interactions and feeder discoveries into account properly predicts that B should arrive first. As such, here we use an NBDA variant that incorporates dynamic, time-varying networks (see ref. [26] for additional information). Specifically, networks updated when the next individual to become informed was last observed within the hive prior to arriving at the respective feeder. For events in which an individual left the hive prior to other naïve individuals, but arrived at the feeder after them, we used the latest update time to reflect their link; in other words, networks were not allowed to "rewind".

NBDA can use either the order in which individuals acquire the target behaviour (order-of-acquisition) or the times at which they do so (time-of-acquisition). The latter approach often has more power to detect social transmission, but also makes more stringent assumptions about the shape of the baseline acquisition rate function: i.e., the manner in which the rate of acquisition changes over time[23]. A honeybee forager's daily routine is determined by a complex mix of factors both internal (e.g., a colony's nutritional needs) and external (e.g., resource availability and temperature) to the hive, thus making a constant, or monotonically increasing or decreasing baseline rate function likely inappropriate for our study. Consequently, we opted to employ order-of-acquisition diffusion analysis (OADA), which assumes only that the baseline rate function is the same for all individuals. In order to increase our power to detect social transmission, we used an OADA variant that is sensitive to differences in learning rate across groups. This was done by including all four colonies in the same diffusion, taking as data the recruitment/reactivation order across all colonies, but setting all among-colony network connections to zero (see ref. [41] for additional information).

We assessed the ability of three network types to predict feeder arrival order for both reactivation and recruitment: (i) dance-following, (ii) trophallaxis and (iii) antennal body contact. All networks were directed, with links travelling from individuals collecting sucrose from the FULL feeder towards individuals either to be recruited (EMPTY group) or to be reactivated to their familiar feeder (FULL group). Connections in the dance-following network were weighted according to the number of waggle runs followed, as waggle runs contain the components known to encode spatial information[1]. For both the trophallaxis and antennal contact networks, two network variants were considered: network connections were either weighted by interaction duration (s) or by the number of unique interaction events (i.e., ignoring the duration of each event).

We included three additional predictor variables in the NBDAs to test for potential influences on asocial and social learning rates: (i) trial ID, (ii) the number of days of foraging experience at a feeder's 200 m location (mean-centred; reactivation diffusion only) and (iii) the number of trips to the EMPTY feeder before locating the FULL feeder (mean-centred; recruitment diffusion only; time-varying variable that updated with the networks). The impact of these variables on asocial and social learning rates were modelled as linear effects on the log scale[23,41]; for further information, see Supplementary Note 2.

Across all four colonies, 91 individuals reactivated to the FULL feeder and 56 individuals were recruited to the FULL feeder during the 2-h trial period (Table 1). For the recruitment NBDA, any individual that had been trained to the FULL feeder was treated as informed from the start of the trial, but only began

transmitting to nestmates once it began collecting from the FULL feeder during the trial. For the reactivation NBDA, some individuals were present at the FULL feeder from the start of the trial or shortly thereafter, or were not observed within the hive prior to arriving at the feeder ($n = 24$). These individuals were excluded as potential learners in the reactivation NBDA—though they were still permitted to transmit information after arriving at the FULL feeder—in order to restrict our analysis to the subset of individuals that could have engaged in our focal transmission pathways (whether or not they actually did so). In addition, ten marked individuals were recruited to the FULL feeder without having been trained to either feeder location; these individuals were also included in the reactivation NBDA as previously informed demonstrators, but not as potential learners.

We employed a multimodel inferencing approach[34] in which every NBDA model for all combinations of the following parameters was constructed. Models could include up to three separate network types: dance following, trophallaxis and/or antennal body contact. For both the trophallaxis and antennal contact networks, variants were considered in which network edges were weighted either by the number of interactions or their duration. Only one variant for a given network type was included within a model—e.g., a model that included "trophallaxis (duration)" did not also include "trophallaxis (number of interactions)". Models were also considered in which social transmission rates (s) were allowed to vary across the trophallaxis and antennal contact networks, or were constrained to be equal. For the reactivation NBDA, two additional predictors were considered: trial ID (1/2/3/4), and the number of days of foraging experience at the feeder's 200 m location; the former was treated as a factor, and the latter as continuous. For the recruitment NBDA, additional predictor variables included: trial ID and the number of visits to the EMPTY feeder during the trial, with the latter treated as continuous. We considered models in which each predictor could either influence asocial learning only, social learning only, or an "unconstrained" model that estimated effects on asocial and social learning as independent parameters[41]. We also included models in which only asocial learning was possible ($s = 0$), as well as models in which social transmission was assumed to operate homogeneously within colonies. This latter option assesses the possibility that our measured networks failed to reflect the actual transmission pathways at work[41].

Homogeneous networks were constructed by forming connections between each individual when it first returned to the nest after visiting the FULL feeder and every other colony member that had yet to be reactivated or recruited (depending on the diffusion), save for those individuals who had left the hive—either to return to their familiar feeder or because they had been successfully recruited—prior to the former's return. For informed individuals whose first return to the hive was not observed ($n = 6$), we assumed they began transmitting when they first arrived at the FULL feeder instead. Constructing the homogeneous networks in this way attempts to account for all possible means of within-nest social information transfer, thereby serving as an appropriate "null" model relative to our measured networks.

For each model, we obtained its Akaike information criterion corrected for sample size (AICc) and used these to calculate Akaike weights ($w_i$) indicating the relative support for a model given the data[34]. Overall support for each network type, predictor variable or for models in which only asocial learning was permitted was calculated by summing $w_i$ across each model in the full model set that included the element of interest. Model averaging was used to obtain parameter estimates[34]. For model-averaging, we excluded models that included the homogeneous networks, as these models greatly skewed model-averaged estimates yet received virtually no support ($\Sigma w_i < 0.001$). For the trophallaxis network, whichever variant (in terms of whether connections were weighted according to the number of interactions or their duration) that received less support was also excluded for model-averaging; this was also done for the antennation network. Finally, we further excluded any models that failed to satisfactorily converge. Note that for non-converged models, AICc (and by extension, Akaike weights) can still be trusted. This is because convergence failure arises when a range of values for one or more parameters have a very similar log-likelihood; this makes it difficult to find the maximum likelihood estimators, but this also means that the log-likelihood returned for a non-converged model, and thus the AICc, will be accurate to several decimal places.

Standard errors for social transmission parameters, s and other predictors could not be reliably obtained due to highly asymmetrical profile likelihoods for one or more parameters. This also makes standard errors a misleading measure of precision, e.g., for s parameters there can often be a large amount of information about the lower limit but very little information about the upper limit. Therefore, we derived 95% confidence intervals for parameters using profile likelihood techniques[67] based on the best predictive model that included a given parameter.

To evaluate whether the effects of social transmission on recruitment and reactivation may have depended on interactions between networks, we adopted the following procedure. For each NBDA, we first constructed a model that included all variables for which $\Sigma w_i > 0.5$. For instance, in the Recruitment NBDA, this model included only the dance network and an effect of the number of return visits to the EMPTY feeder on the rate of social transmission. This model was then compared to alternative models that included that same set of variables along with one or more additional network effects; these could include both interactions between networks, as well as non-interactive network effects. Interaction effects were estimated as the social transmission parameter for a network that was obtained by multiplying together the adjacency matrices for the networks of interest. Models that included an interaction effect also included both of the constituent networks making up that interaction

$\left(\text{e.g.,} s_{\text{Dance}} + s_{\text{Trophallaxis}} + s_{\text{Dance∗Trophallaxis}}\right)$. Model support was compared on the basis of AICc.

All analyses were conducted in R (https://www.r-project.org/) using the NBDA package ver. 0.8.4[68].

**Reporting summary**. Further information on research design is available in the Nature Research Reporting Summary linked to this article.

## Data availability

Data published in this paper are provided in Supplementary Data 1. The associated raw data for Supplementary Fig. 1 is provided in the Source Data file.

## Code availability

The network-based diffusion analyses reported in this paper can be carried out using the NBDA package for R available at: https://github.com/whoppitt/NBDA/. R scripts to obtain all values reported in both this paper and the Supplementary Information are provided in the Supplementary Software.

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

## Acknowledgements

We would like to thank Miranda Burke for her assistance in carrying out the experimental work. This research has received funding from the European Research Council (ERC) under the European Union's Horizon 2020 research and innovation programme (grant number 638873).

## Author contributions

M.J.H. and E.L. conceived the study and designed the experiment. M.J.H. collected the data. M.J.H. and W.H. analysed the data. M.J.H., W.H. and E.L. discussed the results and wrote the paper.

## Competing interests

The authors declare no competing interests.
