## [Peer Review File · Nature Communications]

Reviewers' Comments:

Reviewer #1:

Remarks to the Author:

This manuscript presents a very interesting experimental study on waggle dances of honey bees. The main focus lies in assessing the relative importance of alternative social information sources for (1) locating novel feeding sites and (2) re-activating known feeding sites. The results show that the discovery of novel feeding sites is mainly mediated by dances and re-activation of known feeding sites is mainly mediated by olfactory information, specifically by antennation.

The conceptual framework and the statistical analyses are both based on communication networks. I think this technically a valid approach. However, the analyses and the manuscript as a whole could be simplified a lot by removing all or most parts related to networks. On the conceptual side I do not see any benefit in talking about "communication networks" instead of simply referring to different "information sources".

The statistical analyses were conducted by applying network-based diffusion analysis (NBDA). This application is technically correct. However, the data that has been collected in this study is so detailed that the involvement of networks becomes unnecessary. To understand this point it is useful to consider the original purpose of the NBDA. Essentially, NBDA had been developed to deal with incomplete information on social transmission opportunities. Usually this information is very difficult to collect and direct data on social transmission opportunities is not available. To deal with this problem NBDA essentially estimates the missing information. Specifically, NBDA combines information on social behavior (in form of networks) and individual trait acquisition to estimate individuals-specific changes in social transmission opportunities over time. In the current study it is not necessary anymore to estimate social transmission opportunities, because these opportunities are directly observed and recorded (e.g. the number of dances). The application of NBDA in this case means that information on social transmission opportunities is first put into a network, and then the same information is extracted again from the network in order to fit social transmission models. This whole process could be simplified by leaving out the network part and directly using recorded information on social transmission opportunities (number of dances etc.) for fitting alternative transmission models. As an example, the authors first recorded the number of dances for each demonstrator and then summed them up across demonstrators. The same could be achieved by ignoring the identity of the demonstrator from the beginning and just counting the total number of dances for each individual.

Minor points

Title: "in situ dominance" seems to contradict the results because the estimated influence of dancing on re-activating is weaker than the influence of antennation. Furthermore, I suggest adding information in the title that the study was performed on honey bees.

L 106-107: I do not understand why it was surprising that olfactory information use did not predict arrival times at the novel feeder. It seems to me that the design of first experiment eliminates the possibility for using olfactory information in a meaningful way. Thus, this experiment seems to test whether information acquired through dances is used when olfactory information provides no information. It would be useful to explain this, or alternatively explain how it would be mechanistically possible that olfactory information can speed up arrival at the novel feeder in this experimental setting.

Figure 1: The networks seem to be completely uninformative. I suggest removing them or otherwise

explain what information the reader can extract from them.

I wonder whether additional analyses would allow gaining insights on whether and how different information sources are combined. Currently all information sources are assumed to exert independent influences because it is assumed that all effects on the rate of arriving at a feeding site are additive. In principle it would be possible to fit models that include statistical interactions between different information sources, which could represent synergistic effects.

Reviewer #2:

Remarks to the Author:

In "Network-based diffusion analysis reveals in situ dominance of dance communication networks", Hasenjager et al. use the aforesaid mentioned technique to analyze honey bee forager interactions from a field trial to determine if waggle dances, antennation (olfactory info transfer), or trophallaxis (olfactory and reward info transfer) are best at predicting both recruitment, reactivation, or both. They find that recruitment to novel locations strongly is associated with waggle dance interactions and not the other two types of interactions. Reactivation, in contrast, seems to be more associated with antennation.

I liked this paper very much because of its combination of techniques and, most importantly, because it provides support for the importance of the waggle dance in a field-realistic context, something that has been not well understood in the past.

I really have no major concerns about the paper – I think the experiment and the analysis are both sound, and the discussion is thorough. My only medium concern is the scope and content may be less interesting to folks outside of honey bee foraging ecology or, at best, animal behavior in general. It's hard for me to say that because I find the paper extremely interesting and exciting.

I have a few smaller comments that I hope the authors will consider that might help with the general readability of the paper. Perhaps too these comments might help increase its appeal to a broader readership.

1. Title – the title does little to convey what's going on to anyone unfamiliar with NBDA. In fact, having both the NBDA and the "in situ" made me think this was just a modelling paper without a field experiment. We don't even know the organism! The word "network" is used twice, which always feels clunky to me. Consider a new title that is more informative, doesn't overly rely on the analysis method at the expense of other, important info (organism!), and that conveys the fundamental take-home of the paper – that there are contexts in which the dance communication is dominant.

2. Ln 19 – for those of us that work with the waggle dance, the word "direction" has a very specific meaning: the dance communication encodes a distance and a direction to a resource. So when we want to use one word to describe what the dance provides, we usually say "spatial" or "location" (and indeed, the authors also use "spatial", see Ln 272). Consider replacing "directions" with "location information"

3. Ln 25 – remove comma before "because"

4. Ln 38 – very good explanation of NBDA – understandable to someone who hasn't yet heard of it

5. Ln 42 – "trait" – is it more of a behavioral event? "Behavioral trait" generally means something else

to behavioral ecologists.

6. Table 1 – For trials 3 and 4, you have very few recruits to the full feeder. Do you have any thoughts why? If so, can you discuss them in the discussion?

I suspect that, once you get to mid-September, there actually are other foraging resources available, and this might make the bees less likely to be recruited to another site. You talk in your discussion how you train during autumn b/c there's a forage dearth, but your citations are from Seeley and von Frisch, who definitely say that training is easier when nectar is in short supply, but say nothing about the seasonality of forage availability for southern England.

In particular, ivy is a very important autumn nectar flow for your part of the world, as been shown with research from Sussex. Please see (1) Garbuzov & Ratnieks (2013) *Insect Conservation and Diversity* (2) Couvillon et al (2014) *Plos One* for better references re: forage availability in autumn.

7. Fig 1: I found most of figure 1 to be very helpful, and indeed I referred to it many times to make sure I understood the experimental design. The only notable exception was 1e – there's nothing to be gained here at all. What am I looking at? I know what a general network looks like – this adds nothing new. Does the dance versus trophallaxis versus antennation network look different? Is there a better way to represent what you are trying to show?

Consider replacing 1e with something more meaningful to the reader – a picture of a waggle dance versus antennation versus trophallaxis. Someone not directly working with the waggle dance might find that much more useful than an unnamed / unlabeled network.

8. Ln 113 – you throw in “asocial” learning, but a reader might not know what you mean. Can you define social versus asocial (including examples both from the paper and from other model systems)?

9. It struck me that one good example of how the odor issue dominated people's thinking of the waggle dance is the dance controversy (i.e., Adrian Wenner). For fun, read Munz (2005) *Journal of History of Biology*. And you can maybe mention a line in your discussion how, in some ways, your results really put the final nail in the coffin of the idea that odor alone is sufficient and necessary to recruit new foragers to a novel site (Ln 130 it is mentioned, but perhaps you can say something in Ln 303). I think the “dance controversy” underscores how contentious the conversation was about the benefit / use of the dance.

10. Ln 131 – I thought it was interesting that the number of waggle runs followed predicted recruitment order. We know from Seeley that a recruit has a greater chance of following a novel site if the recruit follows more waggle runs. So your results suggest a similar dose-dependent effect of the dance? That strengthens too your main argument. Can that been seen in figure form? Would be nice to have some data graphs -

11. Ln 137 – “differing degrees” – how so? Can you explain?

12. Ln 146 – so the more persistent bees (see Al Toufalia (2013) *Ethology*, it's the ones more determinedly visiting the EMPTY feeder) are less likely to be recruited to new site? That seems very interesting to me.

13. Ln 352 – Again, consider referencing something specific to your area (see comment 6).

14. Ln 376 – clearly your experimental procedure worked, but just a note of interest – 50 uL / L is a

strong scent! We usually used 10 μ L / L – very strong scent can inhibit dancing.

15. Lastly – is there a way to broaden the readership of the paper so it is interesting to people outside of bee foraging ecology and communication / self organization? Persistence of presumably costly behavior when the adaptive benefit has been unclear?

Well done on a very interesting read!

Kind regards,
Dr. Margaret Couvillon

Reviewer #3:
Remarks to the Author:
Review:

This paper describes an elegant experiment that uses social network analysis to disentangle the relative contributions of waggle dances, olfactory cues and nectar donation to 1) the discovery of novel food sites, and 2) rediscovery/return to previously rewarding sites. To do so, the authors apply a modified form of network-based diffusion analysis (NBDA), comparing the fit between the order of arrival of bees to the food sites with social networks built from observations of interacting bees at the hive.

NBDA is a well-established modelling approach to investigating the transmission of information. It is more usually used with a single information network, and I haven't previously seen it used to compare alternative information networks in quite this way (Farine et al. 2015 possibly excepted). However, it seems to be a logical extension of the approach and I can see lots of potential future applications following a similar methodology to disentangle alternative possible information pathways.

In general, I think the paper is well written, of broad interest, and well suited to publication in Nature Communications. I think the first experiment (discovery of new feeder) is excellent with a very clear result, and I only have minor comments requiring clarification. The second experiment (reactivation) is a little harder to understand, and I would appreciate more information/clarity on the experimental design and analysis.

Experiment 1:

1) Could the authors explain why they had an experimental design that involved the paired "empty" / "full" feeders, rather than simply one novel food site that some proportion of individuals had been trained on? At L50 they state it is "context mimicking natural depletion of one patch...followed by discovery of another", but it is not clear to me why it this empty site is necessary at all to help answer the central question, that is, what information individuals use to discover new food sources.

2) Why were there so few recruits to the full feeder for the third and fourth trials? These two trials appear to much fewer bees discovering the feeder (e.g. 4 in the third trial vs. 34 in the second). Does this have any effect on the analysis?

3) Table 1 - I was scratching my head over this for a while to figure it out. I'm not convinced this table is needed in the main text at all, but if the authors want it, can it be simplified? For example, is it really necessary to know the mean air temperatures, given there was likely a lot of variation over the

day?

4) Did the bees arrive to the new food sites in groups, or follow each to food sources outside the hive? Were these possible "out of hive information transfers" controlled for in any way in the NBDA model?

Figure 1 – these are pretty uninformative networks. I would suggest that instead of showing 4 'dancing' networks, you instead show 3 networks for one site as an example: 1 x waggle dance, 1 x trophallaxis, 1x antennal (you could include all the networks for the other three sites in the supplementary). That would allow the reader to compare and contrast the networks derived these different interactions.

Experiment 2:

I find this experiment a little confusing – but if I can attempt to summarise, previous to the trial, the two feeders are equally profitable and provide identical scent cues. During the trial, only one is baited, but all decisions to return to either feeder are recorded as reactivation events.

1) If reactivation to either feeder was included in the analysis, what was the thinking behind only filling one feeder? Why not just record individuals returning to 1 previously rewarding feeder, or even 2 previously rewarding feeders?

2) The authors state 173 that some individuals were already present and waiting at the familiar feeders, having presumably "reactivated themselves" based off their memory, and without any extra social information. This process presumably continued on throughout the trial. Can the authors make it clearer how they accounted for this personal information use operating parallel to the social information? I realise that they included a model that only had asocial effects (L 187), but it doesn't seem to be me to be a sufficient control to assume it is either one or the other – rather having extensive personal information use would presumably be making their results much "messier" and more difficult to interpret. I confess I'm finding it difficult to see how the model could accurately estimate all these information sources (including bees dancing for natural sources as well!), with multiple potentially information users and information providers. More justification/information would be appreciated.

3) The hypothesis stated on line 178-180 is that olfactory and motivation information provided by return successful foragers would be relevant for reactivation to either feeder. Presumably it also follows that waggle dances would only be relevant for the full feeder.

Is it therefore the first prediction that kinds of different information use should lead different numbers of forages presenting at the full and empty feeder? After I read this paragraph I expected the first presented result to the proportion/numbers of individuals returning to each site, with a bias to the full feeder indicating a role for the spatial information provided by the waggle dance. While in L228 the authors include the feeder status in the NBDA model, it seems to me that the raw data is also informative. Can you provide this information?

4) L242 – why were individuals trained to the empty feeder slower to reactivate?

Signed,
Lucy Aplin

Reviewer #1 (Remarks to the Author):

This manuscript presents a very interesting experimental study on waggle dances of honey bees. The main focus lies in assessing the relative importance of alternative social information sources for (1) locating novel feeding sites and (2) re-activating known feeding sites. The results show that the discovery of novel feeding sites is mainly mediated by dances and re-activation of known feeding sites is mainly mediated by olfactory information, specifically by antennation.

The conceptual framework and the statistical analyses are both based on communication networks. I think this technically a valid approach. However, the analyses and the manuscript as a whole could be simplified a lot by removing all or most parts related to networks. On the conceptual side I do not see any benefit in talking about “communication networks” instead of simply referring to different “information sources”.

The statistical analyses were conducted by applying network-based diffusion analysis (NBDA). This application is technically correct. However, the data that has been collected in this study is so detailed that the involvement of networks becomes unnecessary. To understand this point it is useful to consider the original purpose of the NBDA. Essentially, NBDA had been developed to deal with incomplete information on social transmission opportunities. Usually this information is very difficult to collect and direct data on social transmission opportunities is not available. To deal with this problem NBDA essentially estimates the missing information. Specifically, NBDA combines information on social behavior (in form of networks) and individual trait acquisition to estimate individuals-specific changes in social transmission opportunities over time. In the current study it is not necessary anymore to estimate social transmission opportunities, because these opportunities are directly observed and recorded (e.g. the number of dances). The application of NBDA in this case means that information on social transmission opportunities is first put into a network, and then the same information is extracted again from the network in order to fit social transmission models. This whole process could be simplified by leaving out the network part and directly using recorded information on social transmission opportunities (number of dances etc.) for fitting alternative transmission models. As an example, the authors first recorded the number of dances for each demonstrator and then summed them up across demonstrators. The same could be achieved by ignoring the identity of the demonstrator from the beginning and just counting the total number of dances for each individual.

It is technically correct that we could fit our models without necessarily adopting a network-based framework. The model is equivalent when described in the NBDA framework as when described as a Cox model (although the latter would require modification, given that assuming an exponential relationship between total number of interactions and rate of recruitment would be unrealistic in this case). However, we feel that there are several benefits to the network approach. Briefly, these include:

(i) Extendibility, An important secondary objective of our manuscript is to illustrate new possibilities in the use of NBDA to researchers working on a diverse array of study organisms. The NBDA package developed alongside this manuscript allows for the incorporation of multiple network types in a single model, as well as dynamic, time-varying networks. Until now, the code to implement these options has not been publicly available. Along those lines, we were pleased by Reviewer 3's comment that these extensions should open up new lines of inquiry for social learning researchers.

(ii) Practicality. Unlike the modified Cox approach, the abovementioned *NBDA* package allows for fitting these models without requiring that we develop custom code, which would limit its potential for application outside our system.

(iii) Avoiding potential confusion by referring to equivalent models differently (e.g., *NBDA* vs some other model) depending on the nature of the network data itself. We now include Supplementary Note 2 that sets out the mathematical formulation of our models and addresses these concerns in greater detail.

Minor points

Title: "in situ dominance" seems to contradict the results because the estimated influence of dancing on re-activating is weaker than the influence of antennation. Furthermore, I suggest adding information in the title that the study was performed on honey bees.

Thank you for pointing this out. We have now adjusted the manuscript title to reflect the context-specificity of the importance of waggle dance communication, as well as actually mentioning our study organism!

L 106-107: I do not understand why it was surprising that olfactory information use did not predict arrival times at the novel feeder. It seems to me that the design of first experiment eliminates the possibility for using olfactory information in a meaningful way. Thus, this experiment seems to test whether information acquired through dances is used when olfactory information provides no information. It would be useful to explain this, or alternatively explain how it would be mechanistically possible that olfactory information can speed up arrival at the novel feeder in this experimental setting.

We have extensively re-written our description of the experimental protocol to improve clarity as to how we envisioned olfactory pathways potentially shaping the recruitment process. It is important to appreciate that bees almost never abandon a feeder unless they directly experience it as currently unrewarding, and they usually return to an empty feeder many times before giving up, even when dances for much better alternatives are taking place in the hive (e.g., Grüter & Ratnieks, 2011, *Anim. Behav.* **81**, 949-954). In other words, our foragers "knew" that the EMPTY feeder was no longer rewarding. Thus, the apparently incongruent scent of that same feeder on successful foragers in the hive provides information about the availability of another patch of the same scent, which may have motivated them to search the field for it.

Odour-mediated recruitment to a new site of the same scent as the old one is a plausible possibility. Experiencing odours in the hive is known to actively motivate foragers to locate any other food sources with the same odour (not just familiar ones); indeed, this issue was the basis for a major challenge to von Frisch's original report of the "dance language" that was the source of significant controversy in the latter half of the 19th century (Wenner & Wells, 1990, *Anatomy of a controversy*). We have added discussion of this (lines 79-86, 221-232) and hope that this is now clearer within the text. However, we also agree that it is possible that any effect of olfactory pathways may become more important when a novel scent is presented at the target feeder, and we now add additional lines (150-151; 278-284) to our original discussion of this point, to make it more explicit. Please also note that some of the experiments we carried out this summer and will analyse over the coming year were designed specifically to explore this question.

Figure 1: The networks seem to be completely uninformative. I suggest removing them or otherwise explain what information the reader can extract from them.

We have replaced the networks originally found in Figure 1e with images of our focal interaction types. We hope that these will be more helpful to readers, particularly those that do not work on honeybees. Please note that we have not yet purchased the commercial license required to publish two of these images, but we will do so if the reviewers and editor agree that the images are suitable and add to the manuscript.

I wonder whether additional analyses would allow gaining insights on whether and how different information sources are combined. Currently all information sources are assumed to exert independent influences because it is assumed that all effects on the rate of arriving at a feeding site are additive. In principle it would be possible to fit models that include statistical interactions between different information sources, which could represent synergistic effects.

Thank you for this suggestion. We agree that the current formulation of NBDA, in which all network effects are assumed to be additive, may not be appropriate for all situations. Indeed, we could envisage a scenario in our own study in which the influence of one type of interaction depends on another—e.g., if attentiveness to a dance (and therefore the likelihood of receiving information) depends on having also engaged in trophallaxis. We now discuss such possibilities in the text (lines 108-113, 197-199), and extend our analysis to evaluate whether interactions between network types might explain our data better than purely additive influences.

To do this, we took the top-ranked model from each analysis and compared it with models in which different network interactions were included. For example, for the Recruitment NBDA, only the dance network and a social effect of the number of return visits to the EMPTY feeder were favoured. If we refer to this model as the ‘supported variable model’, or SVM, we compared the SVM to: (i) the SVM + trophallaxis network; (ii) the SVM + trophallaxis network + trophallaxis*dance network; (iii) the SVM + antennation network; and (iv) the SVM + antennation network + antennation*dance network. This same approach was also used for the Reactivation NBDAs, using the corresponding SVMs. However, we found no evidence for interactive influences on the order of recruitment or reactivation (Supplementary Tables 3, 9, and 13).

Moving forward, we hope to explore additional ways in which interactions between different information sources may be included within the NBDA framework.

Reviewer #2 (Remarks to the Author):

In “Network-based diffusion analysis reveals in situ dominance of dance communication networks”, Hasenjager et al. use the aforesaid mentioned technique to analyze honey bee forager interactions from a field trial to determine if waggle dances, antennation (olfactory info transfer), or trophallaxis (olfactory and reward info transfer) are best at predicting both recruitment, reactivation, or both. They find that recruitment to novel locations strongly is associated with waggle dance interactions and not the other two types of interactions. Reactivation, in contrast, seems to be more associated with antennation.

I liked this paper very much because of its combination of techniques and, most importantly, because it provides support for the importance of the waggle dance in a field-realistic context, something that has been not well understood in the past.

Many thanks for the positive comments, which we were happy to receive

I really have no major concerns about the paper – I think the experiment and the analysis are both sound, and the discussion is thorough. My only medium concern is the scope and content may be less interesting to folks outside of honey bee foraging ecology or, at best, animal behavior in general. It's hard for me to say that because I find the paper extremely interesting and exciting.

Thank you, we are very pleased that you (as a honeybee researcher) find the manuscript interesting. We hope that the review submitted by Reviewer 3 (who does not work with pollinators) highlights that the manuscript is also likely to interest researchers outside of the direct subject area: *"...I can see lots of potential future applications following a similar methodology to disentangle alternative possible information pathways. In general, I think the paper is well written, of broad interest, and well suited to publication in Nature Communications."* To bring this point home, we have highlighted at lines 48-50 and 236-239 that our approach offers a broadly applicable technique for tracking information flow through animal groups.

I have a few smaller comments that I hope the authors will consider that might help with the general readability of the paper. Perhaps too these comments might help increase its appeal to a broader readership.

1. Title – the title does little to convey what's going on to anyone unfamiliar with NBDA. In fact, having both the NBDA and the "in situ" made me think this was just a modelling paper without a field experiment. We don't even know the organism! The word "network" is used twice, which always feels clunky to me. Consider a new title that is more informative, doesn't overly rely on the analysis method at the expense of other, important info (organism!), and that conveys the fundamental take-home of the paper – that there are contexts in which the dance communication is dominant.

We apologize for missing out the study species, and have modified the title (see response to Reviewer 1). In keeping with our secondary objective of exemplifying new developments to the application of NBDA, we feel it is important to keep "network-based diffusion analysis" in the title itself to highlight the interest of this manuscript to researchers outside of the social insect community. Note that we have chosen to emphasise the context-specificity of the dance, because (as shown by Reviewer 1's comment regarding the title) for some sections of our target audience, the key message of our paper is that the dance does sometimes dominate, but for others, the novel message is that there are contexts where it does not.

2. Ln 19 – for those of us that work with the waggle dance, the word "direction" has a very specific meaning: the dance communication encodes a distance and a direction to a resource. So when we want to use one word to describe what the dance provides, we usually say "spatial" or "location" (and indeed, the authors also use "spatial", see Ln 272). Consider replacing "directions" with "location information"

As suggested, we have adjusted our terms when discussing the spatial information provided by the dance (e.g. line 24); direction is only used if specifically referring to the component of the dance that actually conveys directional information.

3. Ln 25 – remove comma before "because"

Thank you for catching this. During revisions, this sentence was rewritten in a way that now requires the comma (lines 28-32).

4. Ln 38 – *very good explanation of NBDA – understandable to someone who hasn't yet heard of it*

Thank you - we are pleased that you found it so.

5. Ln 42 – *“trait” – is it more of a behavioral event? “Behavioral trait” generally means something else to behavioral ecologists.*

Although “behavioural trait” has been commonly used in the past to refer to the behaviours of interest in the social learning literature, we recognize that this terminology may be confusing to readers more familiar with alternative usages. Therefore, we now simply refer to these traits as “behaviours” or “pieces of information” (e.g. lines 46-48), or specifically identify the behavioural pattern of interest (i.e. arrival at the target feeder).

6. Table 1 – *For trials 3 and 4, you have very few recruits to the full feeder. Do you have any thoughts why? If so, can you discuss them in the discussion?*

I suspect that, once you get to mid-September, there actually are other foraging resources available, and this might make the bees less likely to be recruited to another site. You talk in your discussion how you train during autumn b/c there's a forage dearth, but your citations are from Seeley and von Frisch, who definitely say that training is easier when nectar is in short supply, but say nothing about the seasonality of forage availability for southern England.

In particular, ivy is a very important autumn nectar flow for your part of the world, as been shown with research from Sussex. Please see (1) Garbuzov & Ratnieks (2013) Insect Conservation and Diversity (2) Couvillon et al (2014) Plos One for better references re: forage availability in autumn.

We agree with your suggestion that this may be related to the emergence of ivy in mid-September here in southern England. Evidence supporting this view was the high prevalence of dancing for natural sources that occurred in the last two trials (Supplementary Table 4). We now make this clear and cite both suggested references (lines 114-119). Please also see response to Reviewer 3 on this point; briefly, removing these trials from the analysis did not qualitatively change the results.

7. Fig 1: *I found most of figure 1 to be very helpful, and indeed I referred to it many times to make sure I understood the experimental design. The only notable exception was 1e – there's nothing to be gained here at all. What am I looking at? I know what a general network looks like – this adds nothing new. Does the dance versus trophallaxis versus antennation network look different? Is there a better way to represent what you are trying to show?*

Consider replacing 1e with something more meaningful to the reader – a picture of a waggle dance versus antennation versus trophallaxis. Someone not directly working with the waggle dance might find that much more useful than an unnamed / unlabeled network.

We have removed Figure 1e, and replaced it with images depicting the three different behaviours that are the focus of the manuscript. Please note that we have not yet purchased the commercial license needed for two of these figures, but we will do so should the chosen images be considered an appropriate addition to the manuscript.

8. Ln 113 – you throw in “asocial” learning, but a reader might not know what you mean. Can you define social versus asocial (including examples both from the paper and from other model systems)?

We now include lines 121-124 to clarify the specific process encompassed by an “asocial learning only” NBDA model for the recruitment diffusion (see also the captions for Tables 2 & 4). We very much appreciate the need for clarity, but we were reluctant to interrupt the flow of the text here to include examples from other species, and so we have instead aimed to make it clear that we refer simply to learning through independent search. We retain the term “asocial learning” rather than “independent learning” for consistency with the NBDA literature. We also include lines 134-137 to illustrate how the NBDA model inherently includes an asocial learning component, and estimates the acceleratory impact of social learning above and beyond that expected of the former.

9. It struck me that one good example of how the odor issue dominated people’s thinking of the waggle dance is the dance controversy (i.e., Adrian Wenner). For fun, read Munz (2005) *Journal of History of Biology*. And you can maybe mention a line in your discussion how, in some ways, your results really put the final nail in the coffin of the idea that odor alone is sufficient and necessary to recruit new foragers to a novel site (Ln 130 it is mentioned, but perhaps you can say something in Ln 303). I think the “dance controversy” underscores how contentious the conversation was about the benefit / use of the dance.

Thank you for raising the connection to the ‘dance language’ controversy. This is an important point, and we now address it prominently right at the beginning of the discussion (lines 221-232). However, as discussed in our response to Reviewer 1, it may be the case that had a novel scent been used during the recruitment trials, the olfactory pathways may have been relatively more important than in the experiments presented here. As mentioned earlier, we are currently conducting experiments to test this idea, and we discuss it within the text at lines 278-284.

10. Ln 131 – I thought it was interesting that the number of waggle runs followed predicted recruitment order. We know from Seeley that a recruit has a greater chance of following a novel site if the recruit follows more waggle runs. So your results suggest a similar dose-dependent effect of the dance? That strengthens too your main argument. Can that been seen in figure form? Would be nice to have some data graphs –

We agree that this dose-dependent effect strengthens our core argument, and we now emphasize this point in lines 99-102. We also now include Supplementary Figure 1 to illustrate this result graphically, which plots for each recruitment event the total number of waggle runs followed by all individuals and highlights the individual that actually was recruited during that event. The red line connecting these individuals is nearly always above the blue line (which indicates the average number of dances followed by individuals up to that point), which highlights the clear link between the number of waggle runs followed and the likelihood of successful recruitment.

11. Ln 137 – “differing degrees” – how so? Can you explain?

This phrasing was motivated by how these types of predictor variables (asocial and social) were treated in earlier NBDA studies. Originally, non-network predictors could either influence only asocial learning (termed an ‘additive’ model), or could influence both asocial learning and social transmission, but were constrained to do so by the same amount (termed a ‘multiplicative’ model). The NBDA package developed alongside this manuscript

now allows for models in which these types of predictors can influence asocial learning only, social transmission only, or can impact both asocial and social learning with the effects on each type of learning estimated independently. This latter option is termed an 'unconstrained' model. However, we feel this point would be lost on most readers that would be unfamiliar with the history of NBDA models. Thus, we have rephrased lines 154-156. In addition, we discuss this distinction and provide the mathematical formulation that allows for the unconstrained model in Supplementary Note 2.

12. Ln 146 – *so the more persistent bees (see Al Toufalia (2013) Ethology, it's the ones more determinedly visiting the EMPTY feeder) are less likely to be recruited to new site? That seems very interesting to me.*

Yes- and we now highlight this in lines 167-170.

13. Ln 352 – *Again, consider referencing something specific to your area (see comment 6).*

See response to point (6) above

14. Ln 376 – *clearly your experimental procedure worked, but just a note of interest – 50 uL / L is a strong scent! We usually used 10 uL / L – very strong scent can inhibit dancing.*

Thank you- we will consider this for future projects. We now provide citations to the publications upon which we based our choice of concentration in lines 333-334.

15. *Lastly – is there a way to broaden the readership of the paper so it is interesting to people outside of bee foraging ecology and communication / self organization? Persistence of presumably costly behavior when the adaptive benefit has been unclear?*

As stated above (in response to your earlier comment about the scope of the manuscript), it is our view that although the specific results of the manuscript may be of primary interest to social insect and collective behaviour researchers, the methods used here will be of great interest to researchers working on social learning in a diverse array of taxa. We feel this point is made especially well by Reviewer 3's comments to that effect. However, we have attempted to make this point more explicitly in the paper itself, in lines 48-50 and 236-239.

Well done on a very interesting read!

*Kind regards,
Dr. Margaret Couvillon*

Thank you, and thanks for the time that you invested in reviewing our manuscript.

Reviewer #3 (Remarks to the Author):

Review:

This paper describes an elegant experiment that uses social network analysis to disentangle the relative contributions of waggle dances, olfactory cues and nectar donation to 1) the discovery of novel food sites, and 2) rediscovery/return to previously rewarding sites. To do so, the authors apply a modified form of network-based diffusion analysis (NBDA), comparing the fit between the order of arrival of bees to the food sites with social networks built from observations of interacting bees at

the hive.

NBDA is a well-established modelling approach to investigating the transmission of information. It is more usually used with a single information network, and I haven't previously seen it used to compare alternative information networks in quite this way (Farine et al. 2015 possibly excepted). However, it seems to be a logical extension of the approach and I can see lots of potential future applications following a similar methodology to disentangle alternative possible information pathways.

In general, I think the paper is well written, of broad interest, and well suited to publication in Nature Communications. I think the first experiment (discovery of new feeder) is excellent with a very clear result, and I only have minor comments requiring clarification. The second experiment (reactivation) is a little harder to understand, and I would appreciate more information/clarity on the experimental design and analysis.

We were happy to receive your positive response, and pleased that you (as a social learning researcher) think the extensions to NBDA we present here will enable new lines of investigation. Prompted by your comments, we have greatly revised our presentation of the reactivation component of the manuscript. We hope these changes have succeeded in making the experimental design and interpretation of this analysis much clearer and more intuitive.

Experiment 1:

1) Could the authors explain why they had an experimental design that involved the paired "empty" / "full" feeders, rather than simply one novel food site that some proportion of individuals had been trained on? At L50 they state it is "context mimicking natural depletion of one patch...followed by discovery of another", but it is not clear to me why it this empty site is necessary at all to help answer the central question, that is, what information individuals use to discover new food sources.

We apologise for the lack of clarity here, and have extensively rewritten our descriptions of the protocol in response. You are correct that in principle, the question of what information sources are important in guiding bees to new food sources could be answered using a single feeder. However, this is a problem for two reasons:

- (a) It would entail marking all or nearly all the foragers in the colony (1000-2000 bees in each case). We had in fact attempted this very thing before deciding it was not feasible for our set up, partly because such major disturbance can lead to nest abandonment via swarming.
- (b) It would require that we wait for natural food sources to deplete. Foragers rarely abandon a known food source, even when there are many dances for alternatives in the hive, until it no longer provides food. Our design allowed us to control when this would happen, and thus to ensure that our experiment could be replicated 4 times within the field season.

The dual feeder design used here was adapted from previous work by Christoph Grüter and colleagues (e.g., Grüter et al., 2008, *Proc. R. Soc. B*, **275**, 1321-1327; Grüter & Ratnieks, 2011, *Anim. Behav.* **81**, 949-954). We now explicitly state this rationale in lines 75-77, and 346-348.

2) Why were there so few recruits to the full feeder for the third and fourth trials? These two trials appear to much fewer bees discovering the feeder (e.g. 4 in the third trial vs. 34 in the second). Does

this have any effect on the analysis?

As mentioned above in our response to Reviewer 2, the most likely explanation for this drop in recruitment is the ivy bloom that begins in early autumn in our local area (refs added at line 117). This meant that the probability of our focal bees encountering dances for alternative (natural) food sources was most likely higher for these foragers. However, omission of these latter trials from the analysis does not qualitatively change our findings: the dance network still received very strong support and was estimated to account for 96.5% of recruitment events. As in the main analysis, the only other variable that was strongly supported was an effect of the number of revisits to the EMPTY feeder on the rate of social transmission. The results of this analysis are provided in Supplementary Tables 5-7, and are highlighted in the manuscript at lines 117-119.

3) Table 1 - I was scratching my head over this for a while to figure it out. I'm not convinced this table is needed in the main text at all, but if the authors want it, can it be simplified? For example, is it really necessary to know the mean air temperatures, given there was likely a lot of variation over the day?

We have simplified Table 1 to remove the mean air temperature and scents used during the trials; this latter information has been relocated to line 339 in the Methods. However, we feel that Table 1 does provide useful information that should be included in the main text—i.e., the number of foragers trained, reactivated, and recruited during each trial.

4) Did the bees arrive to the new food sites in groups, or follow each to food sources outside the hive? Were these possible "out of hive information transfers" controlled for in any way in the NBDA model?

There is indeed evidence to suggest that honeybees can rely on out-of-hive information sources to localize a foraging resource (in particular, olfactory cues). However, these effects only occur once an individual has approached very near to the foraging site; despite extensive research on this question by Karl von Frisch and colleagues, there is no evidence to indicate that honeybees follow one another from the hive to the feeding site itself. Note that our feeders were situated 200 m from the hive, and could not be seen from the hive due to trees, other vegetation, natural slopes and buildings. Thus, our analysis sought to decompose the primary within-hive information pathways that may have been important to the bees. Simulations presented in Hoppitt (2017, *Phil. Trans. R. Soc. B*, **372**, 20160418) suggest that if these out-of-nest transfers impacted our findings in any way, the likely result would be to make our estimated transmission rates more conservative. These issues are now addressed in lines 374-379.

Figure 1 – these are pretty uninformative networks. I would suggest that instead of showing 4 'dancing' networks, you instead show 3 networks for one site as an example: 1 x waggle dance, 1 x trophallaxis, 1x antennal (you could include all the networks for the other three sites in the supplementary). That would allow the reader to compare and contrast the networks derived these different interactions.

Thank you for the suggestion. We agree that Figure 1e in its original form did not add as much to the manuscript as we had hoped. At the suggestion of Reviewer 1, we have replaced these networks with images depicting the focal interaction types on which our networks were based. We hope that these will be of greater use to readers, particularly those that are not as familiar with the honeybee system.

Experiment 2:

I find this experiment a little confusing – but if I can attempt to summarise, previous to the trial, the two feeders are equally profitable and provide identical scent cues. During the trial, only one is baited, but all decisions to return to either feeder are recorded as reactivation events.

1) If reactivation to either feeder was included in the analysis, what was the thinking behind only filling one feeder? Why not just record individuals returning to 1 previously rewarding feeder, or even 2 previously rewarding feeders?

We have modified the text at lines 175-181 and 346-355 to highlight that “reactivation” and “recruitment” were examined simultaneously within each trial. We also explain this more clearly now in the legends to Figure 1 and Table 1. Your summary was otherwise correct, in that only the FULL feeder was baited, but all decisions to return to either feeder were recorded as reactivation events.

However, we agree that this experiment is not as intuitive as the Recruitment NBDA. Your comments prompted us to reconsider our analysis of these findings. In particular, we became concerned that the EMPTY and the FULL group received somewhat differing information and yet were analysed in the same way. While both groups received similar trophallactic/antennal information, for the FULL group, the dances indicated their known feeder, but for the EMPTY group, they indicated a different feeder. Originally, we had done this because of our interest in the dance’s motivational role (*cf* location-indicating role), but with hindsight we could see that analysing the two groups together could be misinformative.

We thus decided to focus our reactivation NBDA on the FULL group, in order to permit direct comparison with the Recruitment NBDA. Like the Recruits, reactivating bees in the FULL group receive dance, trophallactic and antennal information that indicates the full feeder. The only difference from the Recruitment NBDA is that the bees are familiar with the feeder. This arrangement now allows us to introduce the concept of reactivation in greater depth (lines 171-176) and in a manner that is consistent with our Recruitment NBDA. The main message of our manuscript- that the dance is important for recruitment and less so for reactivation- is maintained.

This change means that reactivation by some bees (the EMPTY cohort) became omitted from the main text. We feel that this is justified, because the dance’s role in driving recruitment to a non-indicated feeder is of limited interest (and non-intuitive) to general readers. However, because it is potentially interesting to honeybee researchers, we nonetheless report it in Supplementary Note 1 for completeness.

Please note that, for the same reason, we no longer present the secondary reactivation analysis that incorporated only the first 20 min. This analysis incorporated dances for natural food sources, which (again) do not provide the same spatial information as the dances for the FULL feeder.

2) The authors state 173 that some individuals were already present and waiting at the familiar feeders, having presumably “reactivated themselves” based off their memory, and without any extra social information. This process presumably continued on throughout the trial. Can the authors make it clearer how they accounted for this personal information use operating parallel to the social information? I realise that they included a model that only had asocial effects (L 187), but it doesn’t seem to be me to be a sufficient control to assume it is either one or the other – rather having

extensive personal information use would presumably be making their results much “messier” and more difficult to interpret. I confess I’m finding it difficult to see how the model could accurately estimate all these information sources (including bees dancing for natural sources as well!), with multiple potentially information users and information providers. More justification/information would be appreciated.

You are correct in stating that bees were likely to reactivate on the basis of personal information, in addition to reactivation resulting from social interactions. The NBDA already takes this potential personal information use into account, as social transmission rates are estimated relative to a baseline, asocial learning rate. In other words, if personal information use was the primary driver of reactivation, this would result in a low estimated proportion of reactivation events resulting from social transmission. We now clarify this aspect of the model (lines 134-137; captions to Tables 2 & 4), as well as provide the estimated percentage of reactivations explained by personal information use alone (lines 206-210).

We further agree that it would be helpful to include variable(s) in the model that may capture the likelihood that individuals would rely to a greater extent on personal information for reactivation. We therefore included an additional variable in the Reactivation NBDAs: the number of days that a forager collected from its familiar feeding station, once it had reached its final 200 m location, during training. If this experience is important for reactivation, we would predict that bees with more experience would reactivate faster than bees with less experience. Indeed, this does appear to have been the case for the cohort trained to the FULL feeder (lines 213-216).

3) The hypothesis stated on line 178-180 is that olfactory and motivation information provided by return successful foragers would be relevant for reactivation to either feeder. Presumably it also follows that waggle dances would only be relevant for the full feeder.

Is it therefore the first prediction that kinds of different information use should lead different numbers of forages presenting at the full and empty feeder? After I read this paragraph I expected the first presented result to the proportion/numbers of individuals returning to each site, with a bias to the full feeder indicating a role for the spatial information provided by the waggle dance. While in L228 the authors include the feeder status in the NBDA model, it seems to me that the raw data is also informative. Can you provide this information?

In fact, simple motivation to search for food sources is thought to be a major function of dance-following, even when searching for a different site than that indicated by a followed dance (Grüter et al., 2008, *Proc. R. Soc. B*, **275**, 1321-1327). For example, Biesmeijer and Seeley (2005, *Behav Ecol Sociobiol*, **59**, 133-142) found that the vast majority of of dance-following events that they recorded (75-88%) were more likely to represent a reactivation rather than a recruitment context, especially when bees follow only a few dance circuits at one time. We now emphasize this point at lines 32-34.

However, we recognize that many readers may be unfamiliar with the ability of the dance to reactivate foragers, let alone why we might expect it to reactivate bees to non-indicated sites. This concern is part of what prompted our reanalysis and reorganization of the reactivation NBDAs (see point 1 for Experiment 2). These changes allow us to introduce the concept of reactivation in greater detail (lines 171-176), while placing the less intuitive concept of dances for the FULL feeder potentially reactivating foragers to the EMPTY feeder in the Supplementary material.

We had previously provided the total number of reactivations in Table 1. However, as requested, we now also include the raw data in the text itself (lines 184-186; Supplementary Note 1).

4) L242 – *why were individuals trained to the empty feeder slower to reactivate?*

*Signed,
Lucy Aplin*

This is likely due to the fact that in that particular analysis, the only individuals that reactivated without first engaging in any of our target interaction types were five individuals that reactivated to the FULL feeder. Thus, by the logic of the NBDA model, it is plausible that only individuals that had been trained to the FULL feeder could reactivate solely through a self-initiated process. In reality, it would be difficult to see how this could be true, given that both cohorts had been treated identically up until the morning of the trial. We suspect this finding is simply a result of sampling error and should not be taken too seriously.

However, given our alterations in how the reactivation data was analysed and presented, this earlier result is now irrelevant to the manuscript since reactivation patterns for foragers trained to the EMPTY feeder are analysed separately from those for the FULL cohort.

Reviewers' Comments:

Reviewer #1:

Remarks to the Author:

The authors addressed all my comments and I have no further comments or suggestions.

Reviewer #3:

Remarks to the Author:

I've read the paper through again carefully and reviewed the author's response to my and the other reviewers comments. I'm satisfied with their response and edits, and I find the paper greatly improved because of them. It is an exciting study, with a clear and convincing result. I'm happy to recommend it for publication in its current form.

We would like to once more say that we appreciate the time invested in our manuscript by yourself and all three reviewers. The clarity and quality of our work has been substantially improved by the feedback we received. If there is anything else required from us, please let us know.